# Effects of Environmental and Electric Perturbations on the pKa of Thioredoxin Cysteine 35: A Computational Study

**DOI:** 10.3390/molecules27196454

**Published:** 2022-09-30

**Authors:** Valeria D’Annibale, Donatella Fracassi, Paolo Marracino, Guglielmo D’Inzeo, Marco D’Abramo

**Affiliations:** 1Department of Chemistry, La Sapienza University of Rome, Piazzale Aldo Moro 5, 00185 Rome, Italy; 2Department of Basic and Applied Sciences for Engineering, La Sapienza University of Rome, Via Antonio Scarpa 14, 00161 Rome, Italy; 3Department of Information Engineering, Electronics and Telecommunications, La Sapienza University of Rome, Via Eudossiana 18, 00184 Rome, Italy; 4Rise Technology S.r.l., Lungomare Paolo Toscanelli, 00121 Rome, Italy

**Keywords:** Thioredoxin, PMM, pKa, electric field, molecular communications

## Abstract

Here we present a theoretical-computational study dealing with the evaluation of the pKa of the Cysteine residues in Thioredoxin (TRX) and in its complex with the Thioredoxin-interacting protein (TXNIP). The free energy differences between the anionic and neutral form of the Cysteine 32 and 35 have been evaluated by means of the Perturbed Matrix Method with classical perturbations due to both the environment and an exogenous electric field as provided by Molecular Dynamics (MD) simulations. The evaluation of the free energies allowed us to show that the effect of the perturbing terms is to lower the pKa of Cysteine 32 and Cysteine 35 with respect to the free amino-acid. On the other hand, in the complex TRX-TXNIP, our data show an enhanced stabilization of the neutral reduced form of Cys 35. These results suggest that external electric stimuli higher than 0.02 V/nm can modulate the Cysteine pKa, which can be connected to the tight regulation of the TRX acting as an antioxidant agent.

## 1. Introduction

The evaluation of the protein residue pKa due to the environment is an intriguing challenge from both an experimental and theoretical viewpoint. In this context, the tendency to lose or gain a proton by specific Cysteine residues in Thioredoxin is known to be related to its biological function. Thioredoxin (TRX) takes part in the Thioredoxin system together with its reductase (TrxR) and a molecule of NADPH [1,2]; such a system is an antioxidant and regulatory machinery with a vital role in cell functioning. In fact, it acts as a defence system against general conditions of oxidative stress, which can be due to many stress phenomena (i.e., ROS, virus infections, and radiations exposition [2,3]). All the Thioredoxin enzymes show a highly conserved structure [4,5] with five β strands surrounded by four α helices [5,6]. The active site, composed of the amino acids sequence Cys32-Gly33-Pro34-Cys35 (CXXC motif), is typical of the catalytic site of Thioredoxins of all species [1,5]. The relevance of TRX is due to its ubiquity and its functioning behaviour in the cytoplasmatic environment [1,7], where its role is to mediate cellular responses under stress conditions. TRX activities are regulated by many proteins, in particular by its cytoplasmatic inhibitor TXNIP (Thioredoxin interacting protein). An overexpression of TXNIP during stress conditions, such as ROS oxidative environment, heat shock phenomena, UV, and γ radiation [3,8,9], reduces TRX activity and induces cell death. Active site Cys 32 of TRX is directly involved in the covalent bond with Cys 247 of TXNIP. This bond undergoes a thiol/disulphide exchange, by the action of the buried Cys 35, playing the role of resolving Cysteine (as it is called the Cysteine responsible for nucleophilic attack on Cysteine 32) [10,11,12], after deprotonation. It was already proved that, under a high concentration of ROS, the disulphide bridge that links TRX to its inhibitor is lost, due to the formation of thiosulfinates species that are able to react rapidly with other available thiolate species. Such behaviour enables the TRX antioxidant activity, leading to interaction with many target proteins and nuclear transcription factors [6]. The main aim of this work is to evaluate, with a theoretical–computational hybrid quantum/classical methodology, the pKa values of Cys 35 for TRX alone or bound to TXNIP. The buried position of Cys 35 (see Figure 1) renders the experimental definition of its pKa value extremely difficult [5,13,14,15] and, in this context, the Perturbed Matrix Method (PMM) [16,17,18,19] seems a convenient tool to investigate such a property. Moreover, in order to consider other typical stress factors within the same simulation protocols, the pKa was also calculated in presence of exogenous electric fields as well as at increasing temperatures.

## 2. Materials and Methods

### 2.1. The Perturbed Matrix Method

The Perturbed Matrix Method [16,17,18,19], is a hybrid Quantum Mechanics/Molecular Mechanics (QM/MM) approach [20,21] that combines, with a low computational cost, an ab initio accurate description of the system region of interest—defined as Quantum Center (QC)—with a classical treatment of the surrounding environment, as provided by Molecular Dynamics (MD) simulations. The PMM-MD method has already demonstrated its validity for the characterization and modelling of biological systems [18,19,22,23,24,25]. In this work, it is used for the description of the deprotonation reaction and the calculation of pKa of Cysteine residues in a protein environment. Here, on the basis of literature data [18], the lateral chain of Cysteine (defined as a molecule of methanethiol) was considered as QC. The protein and solvent environment provides the perturbing electric field acting on the selected QC.

In this context, the electronic Hamiltonian operator, $\hat{H}$, is made of two contributions: one unperturbed, ${\hat{H}}^{0}$, defined by the gas-phase properties, and one given by a perturbation term, $\hat{V}$, defined in Equation (Equation 2).
(1)$$\hat{H}={\hat{H}}^{0}+\hat{V}$$

The ${\hat{H}}^{0}$ term is the isolated QC unperturbed Hamiltonian, while $\hat{V}$ is the perturbation operator, that can be obtained *via* a multipolar expansion centred in the QC centre of mass, $r_{0}$:(2)$$\hat{V}≅\sum_{j}\left[V\left(r_{0}\right)-E\left(r_{0}\right)\cdot \left(r_{j}-r_{0}\right)+...\right]q_{j}$$

The j-index refers to all the QC nuclei and electrons, while $q_{j}$ is the charge for each *j*th particle, and $r_{j}$ the corresponding distance from the centre of mass. The V term is the electrostatic potential and E is the electric field, furnished by the perturbing environment. In this study, a more recent version of the PMM approach, with higher order terms derived from the expansion of the perturbation operator around each atom of the QC (atom-based expansion) [17], is used. Within such an approach, the perturbation operator, $\hat{V}$, is expanded within each *N*th atomic region around the corresponding atomic centre $R_{N}$ (i.e., the nucleus position of the *N*th atom of the QC), as defined in the following equation:(3)$$\hat{V}≅\sum_{N}\sum_{j}\Omega_{N}\left(r_{j}\right)\left[V\left(R_{N}\right)-E\left(R_{N}\right)\cdot \left(r_{j}-R_{N}\right)+...\right]q_{j}$$
with *j* running over all QC nuclei and electrons, *N* running over all QC atoms, and $\Omega_{N}$ a step function, being null outside and unity inside the *N*th atomic region. The expansion of the perturbing term is used in this work only for the Hamiltonian matrix diagonal elements, whereas the other Hamiltonian matrix elements are obtained by using the QC-based perturbation operator expansion within the dipolar approximation (Equation (Equation 2). For each frame of the molecular dynamics trajectory, the Hamiltonian matrix is constructed and diagonalized, taking into account the instantaneous perturbation of the environment. This provides a continuous trajectory of perturbed eigenvalues and eigenvectors, used here to estimate the QC ground state energy for the protonated and deprotonated residue.

Note that, in cases where an exogenous electric field is present, it is classically imposed to each atom of the MD simulation box and its associated perturbation is included in the environmental electric term E.

### 2.2. Estimates of Helmholtz Free Energy and pKa

The acid/base reaction of deprotonation, considered for the QC, can be expressed as follows:$$\begin{array}{c} {\mathrm{CH}}_{3}\mathrm{SH}⇌{\mathrm{CH}}_{3}S^{-}+H^{+} \end{array}$$

The Helmholtz Free Energy change for the reported reaction is obtained by considering the average of both the protonated and the deprotonated ensemble, each with its own ionic environment. The expression of ΔA used and reported in Equation (Equation 5), is based on the approximation of ΔH, which is the QC-environment whole deprotonation energy change, with the QC perturbed electronic ground state energy difference, $\Delta U_{e}$. This is due to the negligible internal energy change of the environment, being exactly zero when considering the typical MD force field.
(4)$$\begin{array}{ccc} \Delta A & = & -k_{B}T\ln{\langle e^{-\beta \Delta H}\rangle }_{CH_{3}SH}+\Delta A_{deprot}^{ion}=k_{B}T\ln{\langle e^{\beta \Delta H}\rangle }_{CH_{3}S^{-}}-\Delta A_{prot}^{ion}\\ & ≅ & -k_{B}T\ln{\langle e^{\beta \Delta U_{e}}\rangle }_{CH_{3}SH}+\Delta A_{deprot}^{ion}=k_{B}T\ln{\langle e^{\beta \Delta U_{e}}\rangle }_{CH_{3}S^{-}}-\Delta A_{prot}^{ion} \end{array}$$

Furthermore, $\Delta A_{deprot}^{ion}$ is the relaxation free energy for the deprotonated species (i.e., the methanethiolate molecule in the present study) due to ${\mathrm{CH}}_{3}\mathrm{SH}\to {\mathrm{CH}}_{3}S^{-}$ ionic environment transition, while $\Delta A_{prot}^{ion}$ is the relaxation free energy for the protonated species (i.e., the methanethiole molecule), due to the ${\mathrm{CH}}_{3}S^{-}\to {\mathrm{CH}}_{3}\mathrm{SH}$ ionic environment transition. According to this assumption, $-k_{B}T\ln{\langle e^{\beta \Delta U_{e}}\rangle }_{CH_{3}SH}$ and $k_{B}T\ln{\langle e^{\beta \Delta U_{e}}\rangle }_{CH_{3}S^{-}}$ can be considered as the upper and lower bounds of ΔA and considering $\Delta A_{deprot}^{ion}$≈$\Delta A_{prot}^{ion}$ th average of Helmholtz Free Energy change can be written as:(5)$$\Delta A≅\frac{k_{B}T}{2}\;\ln\frac{{\langle e^{-\beta \Delta U_{e}}\rangle }_{CH_{3}S^{-}}}{{\langle e^{\beta \Delta U_{e}}\rangle }_{CH_{3}SH}}$$

The ensemble average is evaluated by means of the PMM-MD procedure, within the two ensembles depicted by MD trajectories, containing the proper number of counterions to make the overall environment neutral.

Finally, for the pKa estimates [18], the following expression was used,
(6)$$pK_{a}=-\log\frac{\left[{\mathrm{CH}}_{3}S^{-}\right]\left[H^{+}\right]}{\left[{\mathrm{CH}}_{3}\mathrm{SH}\right]}=\frac{{\mu_{CH_{3}S^{-}}}^{Ø}+{\mu_{H^{+}}}^{Ø}-\mu_{CH_{3}SH}^{Ø}}{2.303k_{B}T}$$
where $\mu^{Ø}$ is the standard chemical potential for the shown species involved in the reaction, assuming unitary activity coefficients for a diluted solution. In the conditions of constant volume (i.e., as in the NVT MD ensembles), the standard chemical potential variation corresponds to the deprotonation Helmholtz Free Energy change:(7)$$\Delta A+\Delta A_{vib,solv}={\mu_{CH_{3}S^{-}}}^{Ø}-\mu_{CH_{3}SH}^{Ø}$$

In Equation (Equation 8), the electronic contribution to the deprotonation free energy, ΔA, is estimated by means of MD-PMM procedure, while $\Delta A_{vib,solv}$ is the vibrational contribution to the deprotonation free energy. Thus, the pKa can be expressed as:(8)$$pKa≃\frac{\Delta A+{\Delta A_{vib,solv}+\mu_{H^{+}}}^{Ø}}{2.303k_{B}T}$$

The standard chemical potential of the solvated proton is determined by experimental estimates of the solvation free energy of the proton, $\Delta G_{H^{+}}$ = $\Delta \mu_{H^{+}}$, of the gas phase standard reaction free energy, $\Delta G_{{\mathrm{CH}}_{3}\mathrm{SH}\to {\mathrm{CH}}_{3}S^{-}+H^{+}}^{gas}$ and of the computed unperturbed QC electronic ground state energy difference, $\Delta U_{el}^{0}$:(9)$$\mu_{H^{+}}^{Ø}=\Delta \mu_{H^{+}}+\Delta G_{{\mathrm{CH}}_{3}\mathrm{SH}\to {\mathrm{CH}}_{3}S^{-}+H^{+}}^{gas}+k_{B}T\ln\left(k_{B}T\rho^{⌀}/P\right)-\Delta U_{el}^{0}-\Delta A_{v}^{gas}$$
and considering the following approximation:(10)$$\Delta G_{{\mathrm{CH}}_{3}\mathrm{SH}\to {\mathrm{CH}}_{3}S^{-}}≃\Delta U_{el}^{0}+\Delta A_{v}^{gas}+\mu_{H^{+}}^{gas}$$
where $\Delta A_{v}^{gas}$ is the vibrational contribution to the gas-phase deprotonation free energy, with the approximation of $\Delta A_{v}^{gas}$≈$\Delta A_{v}^{sol}$, $\mu_{H^{+}}^{gas}$ is the proton gas-phase standard chemical potential, and $k_{B}T\ln\left(k_{B}T\rho^{Ø}/P\right)$ is the correction for the gas to solution standard state change (with $\rho^{Ø}$ as the solution standard state molecular density, 1 M, and P as the gas-phase standard state pressure, 0.1 MPa), the pKa can be estimated as follows:(11)$$pKa≃\frac{\Delta A+\Delta \mu_{H^{+}}+\Delta G_{{\mathrm{CH}}_{3}\mathrm{SH}\to {\mathrm{CH}}_{3}S^{-}+H^{+}}+k_{B}T\ln\left(k_{B}T\rho^{Ø}/P\right)-\Delta U_{el}^{0}}{2.303k_{B}T}$$

Here, the $\Delta \mu_{H^{+}}$ is obtained by averaging between two experimental estimates, reported in the literature [18,26,27], while $\Delta G_{{\mathrm{CH}}_{3}\mathrm{SH}\to {\mathrm{CH}}_{3}S^{-}+H^{+}}$ was evaluated for methanethiol/methanethiolate according to what is reported in Refs. [18,28].

### 2.3. Principal Components Analysis

The Essential Dynamics (ED) of proteins principal collective motions is a multivariate statistical method, based on the diagonalization of a correlation matrix of the atomic positions [29]. The method is based on the assumption of quasi-harmonic internal motions, applied for characterizing large conformational transitions of protein structures evolving along MD simulations.The Principal Components Analysis (PCA) furnishes a description of the concerted motions associated with collective atomic fluctuations. Generally, the covariance matrix is built on a set of selected atomic positions (i.e., C-α atoms). From it, eigenvectors representative of the system principal motion directions and respective eigenvalues are obtained and used to describe the so-called essential subspace. In this way, by considering one or concatenated MD trajectories, it is possible to analyse variations in the structural conformations of the protein.

### 2.4. Computational Details

The unperturbed gas-phase properties (i.e., electronic energies, dipole moments and atomic charges) were calculated for the ground state by means of Density Functional Theory (DFT), with B3LYP [30,31] as functional and 6-31+G* as basis set. For the geometry optimization, 6-31G* was chosen as basis set, due to the negligible differences in using diffuse functions on geometry optimization for small molecules without heavy atoms [32]. On the other hand, for the calculation of the above-mentioned gas phase properties (i.e., energies, dipole moments and charges), the introduction of diffuse functions was necessary to obtain accurate values of the electronic properties [33] in presence of anionic species, such as sulphide. Therefore, the 6-31+G* basis set was used as it is routinely used in molecules containing sulphur [33,34].

Furthermore, B3LYP was chosen as functional in virtue of its feasibility and widespread use in describing biological systems [14,18,23,25,35,36,37], combining computational efficiency and accuracy [38]. These unperturbed properties (energies, dipole moments, and atomic charges) were calculated for the first six excited states by means of Time-Dependent DFT (TDDFT). All the quantum mechanical calculations were carried out with Gaussian 16 [39] software. MD simulations were performed in a cubic box of water solvent (SPC model) for a time of 100 ns, using a time step of 2 fs. The crystallographic structure of the complex TRX-TXNIP was taken from the Protein Data Bank (pdb code 4LL1) [6] All the MD simulations were performed using the Gromacs software package [40] (version 2020). In order to study the thermal effect on the deprotonation reaction, simulations from 300 to 350 K were performed by increasing the temperature by steps of 10 K. The applied electric field, ranging from 0.02 up to 0.12 V/nm, was used in the MD trajectories making use of the Gromacs electric field option and inserted as an additive term to the perturbing electric field inside the PMM [41]. Mean dipole moments along the MD trajectories were estimated with gmx dipole tool. For the PCA, the gmx covar and gmx anaeig tools were used to calculate the covariance matrix and the projection along the first two eigenvectors, respectively. In this work, the covariance matrix was built for the selected group of C-α atoms.

## 3. Results

### pKa Calculation for Cysteine in Different Environments

The Cysteine pKa may vary according to the different protein environments. That is, charged groups, positive electron density at the N-terminal part of α helices (i.e., the orientation of their macrodipole along helix axis), and the solvent accessibility to enzyme active site, can all in principle affect the Cysteine pKa value [12,42,43,44,45,46].

As reference value, the computed pKa of ≈8.2 for the Cysteine (in its zwitterionic form) in water was taken into account and the results are reported in Table 1. The different perturbing conditions considered in this work are: (i) the presence of Thioredoxin interacting protein (TXNIP), covalently linked to Cys 32; (ii) the role of an exogenous static electric field, and (iii) the role of temperature.

As noted by the analysis of the TRX-TXNIP (TRX natural inhibitor) complex, the presence of the inhibitor involves large conformational changes in TRX, which can affect the pKa of Cys 35 [5]. Such a conformational rearrangement lowers the accessibility of the solvent to the active site (as shown in the hydrogen bond plot between solvent and Cys 35 sulfur atom, Figure 2c, and in the solvent-exposed surface analysis, Figure 2d). In addition, the distance between Cys 35 and Cys 32 is reduced from 1.09 to 0.40 nm, in the presence of TXNIP, thus favouring the disulphide bridge exchange reaction (mean distances shown in Table 2).

Therefore, we applied a Principal Component Analysis, i.e., the Essential Dynamics analysis, on the C-α MD trajectory for both TRX and TRX-TXNIP, with Cys 35 in its deprotonated form. We then projected the trajectories onto the first two eigenvectors obtained via PCA, to compare the conformational space sampled in the two cases. Results show that TRX and TRX-TXNIP both sample similar conformations, but TRX explores a wider area, as reported in Figure 2a.

To evaluate the effect of an external electric field on the acidity of TRX Cysteine 35, we applied static electric fields at increasing amplitude values. The lowest electric field intensity was set to 0.02 V/nm, while the highest intensity was set to 0.12 V/nm, with increasing steps of 0.02 V/nm. Note that in this work we implicitly consider the electric field intensity to fulfil the so-called ‘weak field conditions’, i.e., to produce a linear response in the system. The application of a static electric field at increasing field intensities decreases the pKa values, as reported in Table 3.

To further investigate the effects of external electric fields on our system, we performed the PCA analysis with three different exposure conditions: (i) in absence of any applied electric field; (ii) in presence of an applied field of 0.06 V/nm; (iii) in presence of an applied field of 0.12 V/nm. Results indicated that, by increasing the field intensity, the protein gradually changes its accessible conformational space, as shown by the trajectory projection on the C-α essential subspace (see Figure 3).

On the contrary, the temperature increase does not provide a clear trend on the pKa, suggesting that the effect of the temperature is quite limited (see Figure 4 and Table 4). What emerged from our data is that the Helmholtz Free Energy variations are all statistical fluctuations, included inside the measured error, without providing a significant trend with temperature.

Therefore, the pKa decrease due to the applied electric field is not a thermal effect but is due to its direct effect on the dynamical behaviour of the system.

## 4. Discussion and Conclusions

The methodology proposed in this paper allowed for an initial estimation of the value for the pKa of Cysteine in water, found to be ≈8.2 units. This value matches the experimental data well [4,12,47], confirming the validity of PMM-MD as a pKa estimation method [18]. Due to the difficulty of obtaining reliable absolute values of pKa in complex environments (from both theoretical and experimental estimates), in our work, we defined such results in terms of ΔpKa using Cysteine in water as a reference.

After proving the validity of the PMM methodology for this kind of studies, we estimated the pKa values of Cysteines in the complex environment represented by a solvated protein. Thus, for Cys 32 and Cys 35 in TRX we obtained a ΔpKa of −4.7 for both cases. This result is not only consistent with literature data, i.e., an increased acidity of these residues in comparison with the pKa value of Cysteine in water [5,13,14,43,49,50,51], but allows the estimation, with an easy scheme and at a reduced computational cost, of the acid properties of buried residues in a protein environment, thus overcoming the well known experimental limitations. In fact, literature data show a wide range of pKa values obtained via experimental measurements (i.e., fluorescence, UV absorption, and NMR analysis [5]). This difficulty in determining the exact pKa value is relevant for Cys 32 and to a greater extent for the buried residue of Cys 35. Experimental pKa values for Cys 32 spread from ≈6 to 10, although it is generally accepted that the pKa of this residue is lower than Cysteine in water, due to the proximity to the N-terminal part of the α2-helix, that increases its acidity. The role of Cys 35 deprotonation is determinant for the thiol/disulfide exchange reaction occurring in TRX activity. The thiolate form of this residue, responsible for the ${\mathrm{SN}}_{2}$ nucleophilic attack requires a lower pKa than those of the Cysteine in water. Experimental pKa measurements in a buried protein position, such as that of Cys 35 in the CXXC motif, are even less reliable, and literature data (in a range of 7–14 pKa units) are not particularly relevant for a clear comprehension of the acidic properties of the Cysteine in the active site. For this reason, the importance of our results lies in the possibility to understand the mechanisms at the basis of the increased acidity of these residues, which could be linked to TRX activity in antioxidant defence. In particular, Cys 32 in its deprotonated form is able to interact with many target proteins [3,6], while deprotonated Cys 35 acts as resolving Cysteine [10,11,12] in disulphide bridges exchange. The availability of PMM-MD calculations enabled us to selectively characterize the effect of a molecule (TXNIP) covalently linked as well as of exogenous perturbations represented by an applied electric field or a change in temperature.

### 4.1. Inhibitor (TXNIP) Effect

Given the importance of TRX functioning in cellular antioxidant defence, we investigated the effect of the TXNIP inhibitor on the Cys 35 pKa variation. The conformational structural change in the enzymatic loop closure, where the disulphide bridge exchange reaction occurs, is able to facilitate the covalent bond break between TRX and TXNIP. In terms of ΔpKa, we obtained an increase of 2.3 units in the pKa value, making the residue less acidic than in TRX without inhibitor. This effect might be related to a possible conformational change of the protein, consistent with the effect of a substrate covalently linked to TRX [5]. As discussed below, the closure of the enzymatic loop to solvent access is responsible for a destabilization of the anionic sulphide form of Cys 35. This condition lowers its acidity, thus increasing the pKa (from 3.5 in TRX alone to 5.8 in TRX-TXNIP system).

If we assume that such variation is mainly localized on the active site (where the covalent bond between the two proteins takes place) we can circumscribe the analysis on the C-α of α2-helix (data reported in Appendix A). In fact, from the analysis of the eigenvectors components per atom, for the C-α of TRX (reported in Figure 2b) and of the α2-helix alone (reported in Appendix A), the maximum contribution to the first and second eigenvectors is mainly due to the atoms in the active site. This confirms the initial hypothesis of a conformational distortion localized near the CXXC residues. To quantify such an effect on the active site, we evaluated the mean distance between Cys 32 and Cys 35 of the CXXC motif. The computed values are reported in Table 2. Data highlight the role of the inhibitor in affecting the Cys 32 to Cys 35 distance, which is lowered by a factor of 2 when compared to the TRX in absence of the inhibitor. These data show that in presence of the inhibitor, a conformational closure of the active site is observed.

The analysis of hydrogen bonds between the sulphur atom of Cys 35 and the surrounding solvent molecules, conducted with Vmd [52] Hydrogen Bond tool, showed an effective shift in the hydrogen bond distribution mean value, with an average number of H-bonds, which is higher in the TRX without the inhibitor, as shown in Figure 2c. This result highlights a reduced interaction with solvent molecules. In addition, to further support the idea of a closure for the solvent molecules to interact with the enzymatic site, we estimated the solvent accessible area of the CGPC active site motif. The results, reported in Table 2 and in Figure 2 show a reduction of 1.1 nm${}^{2}$ of the exposed surface when TXNIP is covalently bonded to Cys 32 of TRX. Taken together, these data define the role of the inhibitor in lowering the acidity of Cys 35, which implies a decrease in solvent accessibility to the active site. By doing so, the stabilization of the thiolate anionic form of Cys 35 (and consequently its acidity) is decreased.

### 4.2. Effects Induced by Static Electric Field and Temperature Perturbations

One of the goals of the present manuscript was to study the response of TRX protein (which acts in stress conditions, by activating cellular defence mechanisms [2,3]) in response to a strong exogenous electric field. Making use of the PMM-MD procedure, which essentially considers the external electric field as an additive perturbation to the environmental field, we focused on a specific range of field intensities, i.e., from 0.02 up to 0.12 V/nm. This range was chosen because, from our previous works, we know that electric fields with intensities lower than 0.02 V/nm hardly produce meaningful effects, while electric fields higher than 0.12 V/nm could induce a fast denaturation of the protein [53]. In terms of ΔpKa, our data show a significant catalytic effect of the electric field on the deprotonation of Cys 35, as reported in Table 3, thus activating the TRX thiolate form. To quantify such an effect, we calculated the Δ(ΔpKa), i.e., the difference between the ΔpKa in the perturbed and unperturbed condition. By means of our theoretical-computational procedure, we observed that an increase of the external electric field intensities from 0.02 up to 0.12 V/nm corresponds to a decrease of Δ(ΔpKa) up to −16.4 units. Such a trend is also correlated to the variation of the mean dipole moment of the protein, which increases from 311.93 up to 479.36 Debye (from the evaluation of Pearson coefficient we estimated a relatively high correlation of −0.985 between dipole moments and ΔA).

Furthermore, the presence of the helix α2, with its high dipole influencing the pKa of neighbourhood residues [43,44,46] just upon the active site [6], makes the interaction with the external electric field extremely strong, thus enhancing the sensitivity of Cys 35 pKa to the electric field. We then compared these results to the ones obtained via a thermal perturbation (i.e., changing the temperature from 300 K to 350 K). Our data show that the effect of the temperature is quite limited as the Δ(ΔpKa) maximum variation is −3.4 units. These results are reported in Figure 4 and in Table 4, where the pKa and ΔA variation with respect to the electric field intensity and temperature are reported.

### 4.3. Conclusions

In conclusion, what has emerged from this study is a defined characterization of TRX Cys 35 pKa variation, with the explanation of the role of the natural inhibitor TXNIP in closing the active site to solvent access, reducing thiolate stabilization. By considering the effect of an applied electric field, as an additional source of perturbation, TRX has revealed itself as a molecule susceptible to an exogenous applied field, able to lower its pKa value with it, thus enhancing its ability to use Cys 35 as resolving residue in the reaction of disulphide bridges exchange. On the other hand, temperature variation does not provide a clear effect on the pKa, thus suggesting that the electric field effect is not related to the thermal effect, but is due to a perturbation of the dynamical behaviour of the system in the presence of the electric field.

Therefore, we can conclude that TRX activity, under electric field exposure, is correlated to the field intensity, defining a cellular response pathway, via TRX enzymatic role, that acts in defence of stress conditions. To fully understand this intracellular mechanism, our future aim will be a deeper investigation on the effect of an applied electric field, by focusing on the thiol/disulphide interchange reaction, at the basis of TRX active site catalytic functioning.

## Figures and Tables

**Figure 1 molecules-27-06454-f001:**
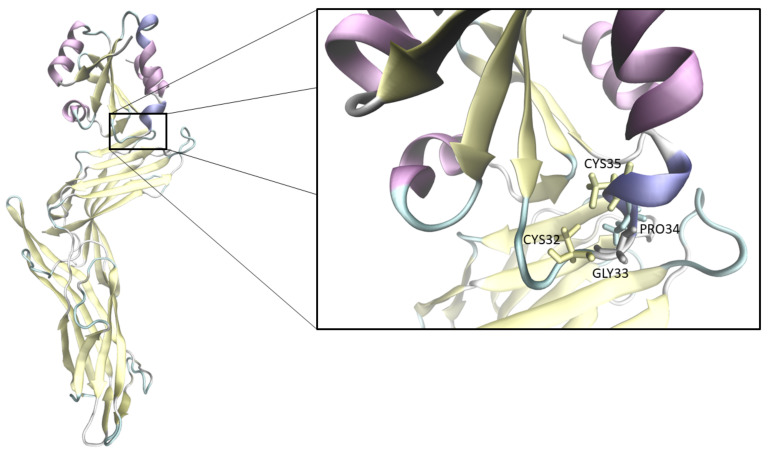
Structure of the complex TRX-TXNIP with a focus on the active site with its CXXC motif.

**Figure 2 molecules-27-06454-f002:**
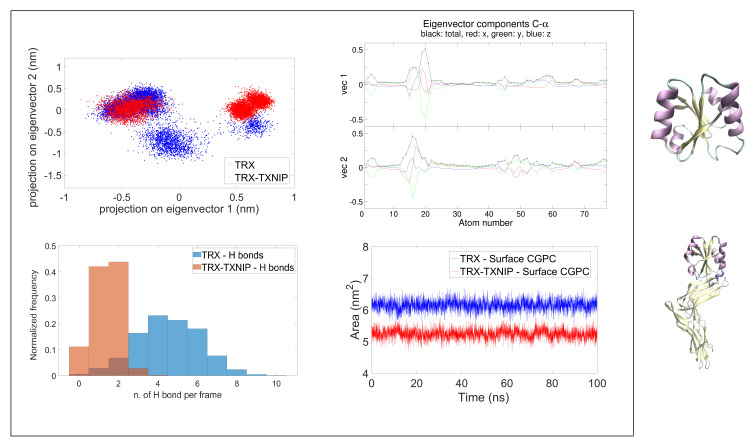
(**a**) PCA analysis on the C-α of TRX and TRX-TXNIP; (**b**) Eigenvector components analysis for the C-α atoms of TRX; (**c**) analysis of h-bonds between Cys 35 and solvent; (**d**) solvent accessible surface of the active site in TRX and in TRX-TXNIP. On the right of the figure, the TRX and TRX-TXNIP complex are shown in cartoon representation.

**Figure 3 molecules-27-06454-f003:**
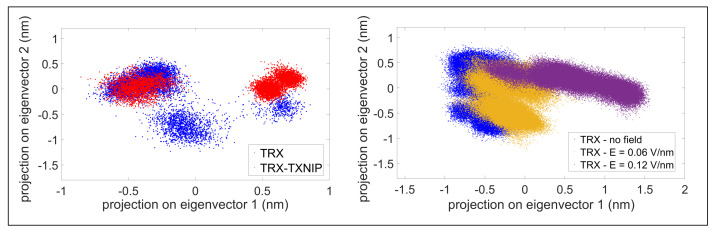
(**left**) PCA comparis on between TRX and TRX-TXNIP (as reported in Figure 2a); (**right**) PCA for the C-α of TRX: no field (blue), E = 0.06 V/nm (yellow), E = 0.12 V/nm (violet).

**Figure 4 molecules-27-06454-f004:**
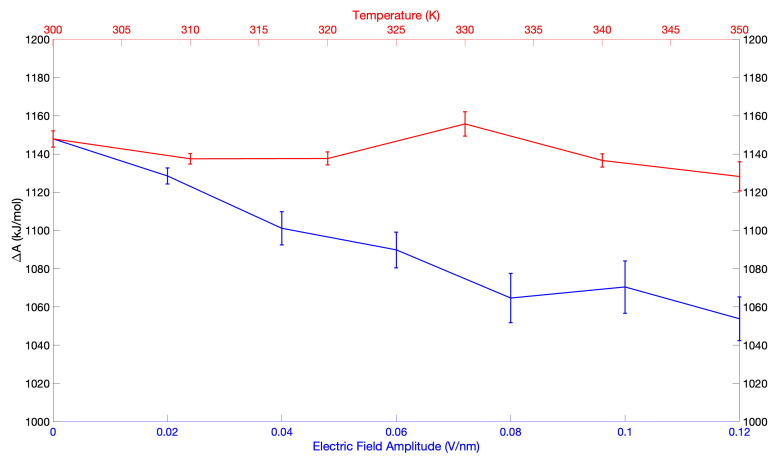
Comparison between the effect of electric field and temperature on Cys 35 in TRX.

**Table 1 molecules-27-06454-t001:** Helmholtz Free Energy change and pKa for Cysteine in water, Cys 32 in TRX, and Cys 35 in TRX and in TRX-TXNIP (always using ${\mathrm{CH}}_{3}\mathrm{SH}$ as QC).

Molecule	ΔA (kJ/mol)	pKa	ΔpKa	pKa (exp. data)
Cysteine in water	1174.9 ± 3.9	8.2	0	8.3–8.8 [4,12,43,47]
Cys 32 in TRX	1147.9 ± 4.1	3.5	−4.7 ± 1.0	5.5–10 [5,13,14,48,49]
Cys 35 in TRX	1147.9 ± 4.3	3.5	−4.7 ± 1.0	7–14 [13,48,49,50,51]
Cys 35 in TRX-TXNIP	1161.2 ± 5.3	5.8	−2.4 ± 1.1	N.A.

**Table 2 molecules-27-06454-t002:** Mean distances along the MD trajectory between Cys 35 and Cys 32, and mean solvent accessible surface of the active site (CGPC), in absence or in presence of TXNIP.

Molecule	Distance C35-C32 (nm)	Active Site (CGPC) Exposed Surface (nm${}^{2}$)
TRX	1.09	6.14
TRX-TXNIP	0.40	5.24

**Table 3 molecules-27-06454-t003:** Effect of a static electric field on the Helmholtz Free Energy change and on the pKa for Cysteine 35 in TRX.

Electric Field (V/nm)	ΔA (kJ/mol)	ΔpKa	Δ(ΔpKa)	Mean Dipole Moment (D)
0.00	1147.9 ± 4.3	−4.7 ± 1.0	0	311.93
0.02	1128.5 ± 4.20	−8.1 ± 0.1	−3.4	332.23
0.04	1101.2 ± 8.7	−12.8 ± 1.7	−8.1	387.08
0.06	1089.8 ± 4.2	−14.8 ± 1.7	−10.1	403.23
0.08	1064.7 ± 12.92	−19.2 ± 2.3	−14.5	470.80
0.10	1070.4 ± 13.7	−17.5 ± 2.5	−12.8	427.02
0.12	1053.8 ± 11.5	−21.1 ± 2.1	−16.4	479.36

**Table 4 molecules-27-06454-t004:** Effect of the temperature on the Helmholtz Free Energy change and pKa for Cysteine 35 in TRX, in the range of 300 to 350 K.

Temperature (K)	ΔA (kJ/mol)	ΔpKa	Δ(ΔpKa)
300	1147.9 ± 4.3	−4.7 ± 1.0	0
310	1137.5 ± 2.7	−6.6 ± 0.8	−1.9
320	1137.7 ± 3.4	−6.7 ± 0.8	−2.0
330	1155.8 ± 6.4	−3.8 ± 1.2	+0.9
340	1136.6 ± 3.5	−6.9 ± 0.8	−2.2
350	1128.3 ± 7.6	−8.1 ± 1.3	−3.4

## Data Availability

Ab initio calculation and geometry optimized structures are open-accessible from figshare online repository (https://figshare.com/account/home#/projects/149882, accessed on 16 September 2022).

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
