# Peer review of "Effects of Environmental and Electric Perturbations on the pKa of Thioredoxin Cysteine 35: A Computational Study"

_molecules, 2022, doi:10.3390/molecules27196454_

Round 1

Reviewer 1 Report

The work by Dr. Annibale and Dr. Fracassi et al. applied a hybrid quantum and classical mechanical calculation to determine the pKa change of Cysteine in different chemical biology environments. The findings and conclusions are interesting. Here, I have listed the following questions to be addressed and discussed in the main text:

1.     The author claims from Fig. 2C and Table 2 that Cys of TRX becomes less accessible to water when binding to the inhibitor (TXNIP), in which case one might expect Cys pKa to decrease. However, as shown in Table 1, Cys pKa increases from 3.5 to 5.8 (being closer to the case of Cys in water pKa=8.2). Does the author has an explanation for this counter-intuitive trend in addition to what they already have written in the main text?

2.     One needs to be careful when stating that temperature has a trivial effect on Cys pKa. The reason is that the protein structure shown in Fig. 1 may not be the the most representative conformation under high temperature, e.g. 350K, in which case, the pKa results calculated for high temperature might be not accurate. 

Minor comments: 

1.     On page 1, line 15, “…in Thioredoxin it is known to be…” should be “…in Thioredoxin is known to be…”

2.     Captions of Fig. 2 are too small to see.

Author Response

 We would like to thank Reviewer 1 for the appreciation of our work

  1. The author claims from Fig. 2C and Table 2 that Cys of TRX becomes less accessible to water when binding to the inhibitor (TXNIP), in which case one might expect Cys pKa to decrease. However, as shown in Table 1, Cys pKa increases from 3.5 to 5.8 (being closer to the case of Cys in water pKa=8.2). Does the author has an explanation for this counter-intuitive trend in addition to what they already have written in the main text?

    1. The limited access of solvent molecules into the enzymatic loop could be responsible for lowering the Cys 35 stability in the thiolate form, thus decreasing the acidity of this residue, which increases its pKa from 3.5 to 5.8.

Accordingly, in subsection 4.1. (Discussion section), a paragraph on the effect of the inhibitor was added.

  1. One needs to be careful when stating that temperature has a trivial effect on Cys pKa. The reason is that the protein structure shown in Fig. 1 may not be the most representative conformation under high temperature, e.g. 350 K, in which case, the pKa results calculated for high temperature might be not accurate.

     2. We performed several MD simulations at different temperatures.

We agree with the referee that the most stable conformations at high temperature can be very different from those at 300 K.

Nonetheless, in the limited time of the MD simulations we observe a partial loss of the structure that is responsible - with the rest of the system - of the observed pKa variation.

Reviewer 2 Report

The author study the  pKa of Thioredoxin Cysteine 35 with Perturbed Matrix Method,  a hybrid Quantum Mechanics/Molecular Mechanics (QM/MM) approach. The core parts are calculated with DFT, while the outside are from classic description. The method have used in pka study in the same group, while the current study extend the work to with/without TXNIP (TRX natural inhibitor). The work looks solid on the methodology.

My major concerning is regarding the conclusion on the effect of thermal.  While the change is relative smaller comparing the full change in electric field.  However given the change on 0.02 V/nm in electric field, change  -3.4 in pka, but the natural regulator TXNIP change only -2.4. In this means, beyond 0.02 in electric field are out of range of biology regulation and less biology significant. In the another side, the effect of thermal are in comparable range. One of the interesting point from thermal is, that the effect peaks around 330, what contributes that? I would suggest move the thermal table into main text and reconsider the biological significance.

Some minor points.

1. It should be possible to unify the PCA for figure 2a and figure 4. Then it could comparing that change on electric field comparing with that from TXNIP. The current way, make it's not comparable. Given the biology relevance for thermal, bring that together could also help.

2. The text and legend are relative small for most of figures, especially for in figure 2 and S1. It's better to increase it for better readability. 

Author Response

ANSWER TO REPORT REVIEW 2

The author study the pKa of Thioredoxin Cysteine 35 with Perturbed Matrix Method,  a hybrid Quantum Mechanics/Molecular Mechanics (QM/MM) approach. The core parts are calculated with DFT, while the outside are from classic description. The method have used in pka study in the same group, while the current study extend the work to with/without TXNIP (TRX natural inhibitor). The work looks solid on the methodology.

We thank the reviewer for his/her appreciation.

My major concerning is regarding the conclusion on the effect of thermal.  While the change is relative smaller comparing the full change in electric field.  However given the change on 0.02 V/nm in electric field, change  -3.4 in pka, but the natural regulator TXNIP change only -2.4. In this means, beyond 0.02 in electric field are out of range of biology regulation and less biology significant. In the another side, the effect of thermal are in comparable range. One of the interesting point from thermal is, that the effect peaks around 330, what contributes that? I would suggest move the thermal table into main text and reconsider the biological significance.

We would like to thank Reviewer 2 for highlight this point.

According to our estimates, the fluctuations of the free energy observed at different temperatures - included the peak at 330 K - are all within the associated statistical errors. This point is now described in the line 210-213 of the revised version of the manuscript.

Following the reviewer suggestion, the table with temperature data has been added to the main text and removed from the SI.

Some minor points.

  1. It should be possible to unify the PCA for figure 2a and figure 4. Then it could comparing that change on electric field comparing with that from TXNIP. The current way, make it's not comparable. Given the biology relevance for thermal, bring that together could also help.

1.Accordingly to the referee suggestion, the PCA analysis reported in Fig. 2 (panel a) has been added to the Fig. 4.

        2. The text and legend are relative small for most of figures, especially for in figure 2 and S1. It's better to increase it for better readability.

2. All the figures are now reorganized and improved in readability.

Reviewer 3 Report

This article is devoted to a theoretical study of environmental and electric perturbations on the pKa of

Thioredoxin Cysteine ​​35. The article is written in a clear and accessible language. The authors performed extensive calculations and studied various effects in molecules. A good theoretical background makes a positive impression when reading this work. In the introduction, the authors quite fully describe the relevance and current state of this topic. Many of the advantages of this work can be increased by finalizing the following points:

1. Abstract can be expanded.

2. It is desirable in this work to indicate more the relationship with experimental data from the literature, as well as to provide more comparisons with other studies from the literature.

3. It is not entirely clear from the text why the authors chose these methods for the study? Why was B3LYP as functional and 6-31+G* as basis set chosen?

4. It is desirable to cite the article: 10.1007/s00894-020-04645-5.

5. In some drawings, the text is not readable. Increase the quality of drawings and unify them.

Author Response

ANSWER TO REPORT REVIEW 3

This article is devoted to a theoretical study of environmental and electric perturbations on the pKa of Thioredoxin Cysteine ​​35. The article is written in a clear and accessible language. The authors performed extensive calculations and studied various effects in molecules. A good theoretical background makes a positive impression when reading this work. In the introduction, the authors quite fully describe the relevance and current state of this topic. Many of the advantages of this work can be increased by finalizing the following points:

We would like to thank the Reviewer for his/her appreciation of our work.

Following our point-to-point response to the referee’s comments:

1. Abstract can be expanded.

1. The abstract was slightly expanded.

2. It is desirable in this work to indicate more the relationship with experimental data from the literature, as well as to provide more comparisons with other studies from the literature.

2. A more detailed comparison between our theoretical data and experimental pKa values reported in literature has been added in the Discussion section (line 229-242)

3. It is not entirely clear from the text why the authors chose these methods for the study? Why was B3LYP as functional and 6-31+G* as basis set chosen?

 3. The explanation of our choice of functional and basis set has been added in the Materials and Methods section (line 142-152) .

4. It is desirable to cite the article: 10.1007/s00894-020-04645-5.

 4. Accordingly to the referee suggestion, the reference has been added to the manuscript bibliography.

5. In some drawings, the text is not readable. Increase the quality of drawings and unify them.

5. All the figures are now reorganized and improved.
